# The Novel Z-Scheme Ternary-Component Ag/AgI/α-MoO_3_ Catalyst with Excellent Visible-Light Photocatalytic Oxidative Desulfurization Performance for Model Fuel

**DOI:** 10.3390/nano9071054

**Published:** 2019-07-23

**Authors:** Yanzhong Zhen, Jie Wang, Feng Fu, Wenhao Fu, Yucang Liang

**Affiliations:** 1Shaanxi Key Laboratory of Chemical Reaction Engineering, School of Chemistry & Chemical Engineering, Yan’an University, Yan’an 716000, Shaanxi, China; 2Institut für Anorganische Chemie, Eberhard Karls Universität Tübingen, Auf der Morgenstelle 18, 72076 Tübingen, Germany

**Keywords:** Z-scheme photocatalyst, Ag/AgI/α-MoO_3_, thiophene, desulfurization

## Abstract

The novel ternary-component Ag/AgI/α-MoO_3_ (AAM) photocatalyst was successfully fabricated by a facile hydrothermal method combined with a charge-induced physical adsorption and photo-reduced deposition technique. X-ray diffraction, scanning/transmission electron microscope, X-ray photoelectron, UV-vis diffuse reflectance, photoluminescence and electrochemical impedance spectroscopy were employed to characterize the composition, morphology, light-harvesting properties and charge transfer character of the as-synthesized catalysts. The ternary-component AAM heterojunctions exhibited an excellent visible-light photocatalytic oxidative desulfurization activity, in which the AAM-35 (35 represents weight percent of AgI in AAM sample) possessed the highest photocatalytic activity of the conversion of 97.5% in 2 h. On the basis of band structure analysis, radical trapping experiments and electron spin resonance (ESR) spectra results, two different catalytic mechanisms were suggested to elucidate how the photogenerated electron-hole pairs can be effectively separated for the enhancement of photocatalytic performance for dual composites AM-35 and ternary composites AAM-35 during the photocatalytic oxidative desulfurization (PODS) of thiophene. This investigation demonstrates that Z-scheme Ag/AgI/α-MoO_3_ will be a promising candidate material for refractory sulfur aromatic pollutant’s removal in fossil fuel.

## 1. Introduction

The serious environmental problems caused by air, water, soil, noise and light pollution have received extensive attention and considerable research interest, especially, air pollution caused by the combustion of sulfur and nitrogen-containing fossil fuel generated SO_x_ and NO_x_, which further leads to water and soil pollution. Desulfurization and denitrification of fossil fuel are becoming a key and challenging topic for controlling air pollution [1,2]. The conventionally industrial hydrodesulfurization (HDS) is not an effective approach to thoroughly/ultra-deeply removing aromatic sulfur compounds (ASCs) in fossil fuel, due to the ASCs with alone electron pairing on the sulfur S atom [3]. Therefore, the development of highly-efficient and environmentally benign techniques is urgent and necessary for removing ASCs in fuels in order to produce ultra-clean fuel [4].

The photocatalytic oxidative desulfurization (PODS), with a lower cost and easy operation under a mild reaction condition is a novel oxidative desulfurization (ODS) technique, which has attracted widespread interest in the treatment of ASCs. For example, during the process of PODS, when Pd/ZrO_2_-chitosan nanocomposite was used as a photocatalyst, thiophene could be photooxidized to the corresponding SO_3_ and CO_2_ under UV-vis light irradiation [5]. At present, in an attempt to remove ASCs in fuel, many photocatalysts including BiVO_4_-based [6,7,8,9], TiO_2_-based [10,11,12,13,14,15,16,17], graphene oxide or carbon nanotubes-based multicomposites [18,19,20] and phosphotungstic acid [21,22] have been used and showed good catalytic performances.

α-MoO_3_ is a chemically stable and nontoxic n-type semiconductor, readily prepared by abundant molybdenum-containing source. Although unique energetic and electrical properties of α-MoO_3_ made it successful as a very promising visible-light driven (VLD) photocatalyst [23,24,25,26], the high recombination rate of photogenerated charge carriers and the low quantum yield greatly restricted its practical application in the photocatalytic field [26]. To solve this problem, enormous efforts have focused on the tunable electronic band structures of α-MoO_3_ by doping and constructing heterojunction to improve photocatalytic activity [27,28]. Some successful photocatalysts, such as Ce-doped MoO_3_ [29], Bi_2_Mo_3_O_12_/MoO_3_ [30], MoS_2_@α-MoO_3_ [31], g-C_3_N_4_/MoO_3_ [32,33] and TiO_2_/α-MoO_3_ [34], have been reported. For semiconductor photocatalysts, it is well known that a suitable conduction and valence band edge potential is critical to facilitate the desired redox reactions in the process of photocatalytic reaction [35]. For α-MoO_3_, the conduction band (CB) has much more positive potential (0.37 eV vs. normal hydrogen electrode (NHE)) than that of O_2_/·O_2_^−^ (−0.33 eV vs. NHE), wherein it is impossible to perform the reduction of O_2_ to ·O_2_^−^ by electrons in the CB. Recently, the artificial Z-scheme photocatalytic system that mimics natural photosynthesis of the green plant has been developed due to superior photocatalytic performance. Such a Z-scheme photocatalytic system containing two independent reduction and oxidation reaction systems not only results in high charge separation efficiency, but also preserves the high redox ability of the photocatalyst, because a Z-scheme pathway of electron transfer can preserve the oxidative holes in the lower VB and reductive electrons in the higher CB. For example, carbon dots (CDs)-modified Z-scheme photocatalyst CDs/g-C_3_N_4_/MoO_3_ exhibited a remarkably enhanced visible-light photocatalytic activity for the degradation of tetracycline [36].

Recently, the silver/silver halides (Ag/AgX, X = Cl, Br, I) photocatalysts have attracted much attention due to their excellent photocatalytic performance and high stability under various light irradiations caused by the surface plasmon resonance effect of metallic silver [37,38]. The constructed Z-scheme heterojunction between Ag/AgX and semiconductor can effectively accelerate the interfacial charge carriers transfer and separation due to Ag nanoparticles as the charge carrier transmission-bridge [39]. Hence, a series of Z-scheme photocatalysts containing Ag/AgX, such as Ag/AgI-δ-Bi_2_O_3_ [40], α-Fe_2_O_3_/Ag/AgX [41], Ag/AgX/AgIO_3_ [42], and g-C_3_N_4_/Ag/BiVO_4_ [43], were successfully fabricated and showed an excellent photocatalytic performance.

The photocatalysis is an environmentally benign technique for removing organic pollutants or dyes/pigments. It is well known that thiophene as a refractory sulfur aromatic pollutant is extremely difficult to remove from fuel, through conventional hydrodesulfurization (HDS) [44]. The previous study revealed that α-MoO_3_-based photocatalysts (α-MoO_3_ nanoblet [45], Ag/α-MoO_3_ [46]) showed good photocatalytic activity for the degradation of thiophene because of their good adsorption performance and the promotion separation of charge carrier by plasmonic metallic Ag nanoparticles.

Combined with the aforementioned analysis, it is desirable to fabricate a Z-scheme heterojunction photocatalyst Ag/AgI/α-MoO_3_ with a boosting photocatalytic activity. To the best of our knowledge, the relative reports about Z-scheme α-MoO_3_-based heterojunctions are quite scarce for the photocatalytic degradation of thiophene.

In the present work, a series of Z-scheme heterojunction photocatalysts Ag/AgI/α-MoO_3_ with a different mass percent of initially formed AgI contents were successfully prepared by a charge-induced physical adsorption and in-situ photo-reduced deposition method. The photocatalytic activities of the as-prepared Ag/AgI/α-MoO_3_ were evaluated by the degradation of thiophene under visible light irradiation. The synergistic effects among metallic Ag, and AgI and α-MoO_3_ in multicomponent AgI/α-MoO_3_ and Ag/AgI/α-MoO_3_ were systematically investigated and combined with UV-Vis diffuse reflectance spectroscopy (UV-Vis-DRS), photoluminescence spectroscopy (PL), electrochemical impedance spectroscopy (EIS), the scavenger trapping experiment and electron spin resonance spectroscopy (ESR) in order to explore the enhanced photocatalytic activity mechanisms towards the decomposition of refractory pollutants. This strategy will create a new avenue for the photocatalytic oxidative desulfurization of fossil fuel in the future.

## 2. Materials and Methods

### 2.1. Sample Preparation

All chemicals with analytical grade were used without further purification and purchased from Sinopharm Chemical Reagent Co. Ltd. (Beijing China) as follows: (NH_4_)_6_Mo_7_O_24_·4H_2_O (99%), HNO_3_ (98%), AgNO_3_ (98.9%), KI (≥ 99.0%), thiophene (99%), ethanol (≥ 99.7%), BaSO_4_, isopropanol alcohol (IPA, ≥ 99.7%), ethylenediamine tetraacetic acid disodium salt (EDTA-2Na, 99%), and 1, 4-benzoquinone (BQ, 98%). Deionized water was used for all reactions, if no any special emphasis is given otherwise.

#### 2.1.1. Preparation of α-MoO_3_ Nanobelt

α-MoO_3_ nanobelt was prepared using a template-free hydrothermal method. In a typical process, (NH_4_)_6_Mo_7_O_24_·4H_2_O (2.0 mmol) in water (10 mL) was stirred for 30 min at ambient temperature to form a clear solution. An aqueous HNO_3_ solution (2.0 M, 10 mL) was then added with stirring and the mixture was continuously stirred for 30 min. Afterwards, the mixture was transferred into a Teflon-lined stainless-steel autoclave with 50 mL capacity and reacted at 180 °C for 8 h in an oven. After the autoclave was naturally cooled down to room temperature, the resulting sample was collected by centrifugation, washed first with water and then absolute ethanol several times, and dried at 80 °C for 6 h under vacuum.

#### 2.1.2. Synthesis of A Series of Ag/AgI/α-MoO_3_ Heterojunctions

Ag/AgI/α-MoO_3_ heterojunctions were fabricated by a charge-induced physical adsorption and in-situ photo-reduced deposition process. To a suspension of as-synthesized α-MoO_3_ (0.5 g) dispersed in ethanol (80 mL), KI (0.1904 g) dissolved in water (10 mL) was added under vigorous stirring. The resulting suspension was sonicated for 20 min, and an appropriate amount of AgNO_3_ dissolved in water was then added dropwise. After stirring for 12 h under a dark condition, the suspension was irradiated for 5 min using a 250 W metal halide lamp at room temperature under stirring. The cut-off filter was used to cut off the light below 420 nm. The precipitate was collected by centrifugation, washed with water and absolute ethanol several times, and dried at 80 °C for 6 h under vacuum. The same procedure was used to prepare as-synthesized a series of samples, AAM-x (x = 15, 25, 35 and 45, which represents weight percent of AgI in sample AAM, for example, AAM-15 means 15 wt % AgI in sample), with different mass percent of initially formed AgI contents in all starting chemicals.

### 2.2. Characterization

X-ray diffraction (XRD) were carried out with a Shimadzu XRD-7000 X-ray diffractometer (Bruker, Karlsruhe, Germany) using Cu Kα radiation (*λ* = 0.15418 nm) at a scanning rate of 2° per minute in the 2*θ* range of 10–80° with an accelerating voltage of 40 kV and an applied current of 30 mA. The surface morphology and microstructures of the sample were observed using a JEOL-6700 (JEOL Ltd., Tokyo, Japan) field emission scanning electron microscope (FE-SEM) and a JEM-2100 (JEOL Ltd., Tokyo, Japan)high-resolution transmission electron microscope (HR-TEM). X-ray photoelectron spectroscopy (XPS) was recorded on a PHI-5400 X-ray photoelectron spectrometer (Ulvac-Phi Inc., Kanagawa, Japan). UV-Vis diffuse reflectance spectra (UV-Vis-DRS) were obtained using a Shimadzu UV-2550 UV-Vis spectrophotometer (SHIMADZU, Kyoto, Japan) in the range of 200–800 nm equipped with an integrating sphere assembly. BaSO_4_ was used as a reflectance standard. Photoluminescence (PL) spectra were measured on an F-4500 fluorescence spectrophotometer (Hitach, Tokyo, Japan) at an excitation of 337 nm. Electrochemical impedance spectroscopy (EIS) was carried out on an CHI660C electrochemical workstation (CH Instruments, Inc., Shanghai, China) using a typical three-electrode configuration, with an ITO glass/sample as a working electrode, a platinum wire as a counter electrode, and a saturated calomel electrode (SCE) in a saturated KCl aqueous solution as a reference electrode. Na_2_SO_4_ (0.5 M) aqueous solution was used as the electrolyte.

### 2.3. Photocatalytic Activity

The photocatalytic activity of as-synthesized samples was evaluated through the degradation of thiophene in a model fluid catalytic cracking (FCC) gasoline. The model FCC gasoline was prepared by dissolving thiophene in *n*-octane, in which the initial sulfur content was about 500 ppm. The reaction system consisted of a 400 W metal halide lamp as the simulated visible-light source equipped with an optical filter (λ ≥ 420 nm) to cut off the light in the ultraviolet region.

In a typical reaction, a certain amount of as-synthesized sample was added into 20 mL of simulated FCC gasoline in a 50 mL quartz tube reactor with water circulation facility, and the mixed solution was then placed in the dark to establish adsorption-desorption equilibrium. Before and during the irradiation, the solution was purged with air as the oxidant. At expected time intervals, 5.0 mL of solution was withdrawn and photocatalysts were separated through centrifugation. The supernatant was then extracted with acetonitrile. The sulfur content of model fuel oil was detected by a WK-2D microcoulometer analyzer with nitrogen as carrier gas, oxygen as combustion gas, an iodide electrode as a reference electrode, and a platinum electrode as a potential electrode. The temperature of vaporization segment is 680 °C and the combustion temperature is 850 °C. The desulfurization ratio for thiophene was calculated by the following formula:$$\eta =(1-\frac{C_{t}}{C_{0}})\times 100\%$$
where *η* stands for desulfurization ratio for thiophene, *C_0_* and *C_t_* are the concentration of initial thiophene and of unconverted thiophene at light irradiation time t, respectively.

Furthermore, trapping experiments were performed to probe the main active species in the photocatalytic process. The experimental apparatus and procedures were identical for all photocatalytic activity tests, but different types of scavengers, including isopropanol alcohol (IPA) for hydroxyl radical (OH), ethylenediamine tetraacetic acid disodium salt (EDTA-2Na) for hole (h^+^), and 1, 4-benzoquinone (BQ) for superoxide radical (O_2_^−^), were added into the simulated FCC gasoline.

## 3. Results

### 3.1. Formation and Characterization of Ag/AgI/α-MoO_3_ Heterojunctions

As shown in Scheme 1, AgI/α-MoO_3_ heterojunctions were fabricated by homogeneous adsorption of iodide ions on the surface of α-MoO_3_ nanobelts in an ethanol solution via electrostatic interaction. The surface of α-MoO_3_ nanobelts is positively charged which can firstly absorb anionic I^−^ ions, then the attached I^−^ further reacts with Ag^+^ ions to fabricate the AgI/α-MoO_3_ heterostructure [24]. Under visible-light irradiation, the AgI on α-MoO_3_ surface was partially converted into metallic Ag nanoparticles and therefore resulted in the formation of a novel ternary Ag/AgI/α-MoO_3_ (AAM) heterojunction.

For AAM heterojunctions with different mass percent of initially formed AgI, the XRD patterns (Figure 1) clearly indicated characteristic diffraction peaks of as-made pure α-MoO_3_ nanobelts with orthorhombic *Pbnm* symmetry (JCPDS No. 05-0508) and *γ*-AgI with face-centered cubic *F-43m* symmetry (JCPDS No. 09-0399). No featuring diffraction peaks of Ag nanoparticles were observed due to the low content of metallic Ag nanoparticles and the overlapping of α-MoO_3_ diffraction peaks. Note that the intensities of diffraction peaks of AgI in AAM gradually increased with increasing AgI loading, and the corresponding value of I_AgI_ (111)/I_MoO3_ (040) also increased. No other diffraction peaks were detected, implying the high purity of the final products obtained. These findings are in accordance with the expected composites.

To investigate the composition of the AAM samples and chemical states of every element, X-ray photoelectron spectroscopy (XPS) analysis was performed and showed in Figure 2. All main peaks corresponded to O, Mo, Ag and I were detected (Appendix A), and the C signal derived from the residual/adventitious carbon of the XPS instrument. A broad O 1s spectrum (Figure 2a) can be decomposed as three peaks at 530.8, 531.7 and 532.6 eV, corresponding to the Mo-O, Mo-OH, and surface adsorbed oxygen, respectively [21,22,23]. The peaks at 232.6 and 235.8 eV in Figure 2b respectively belong to Mo 3d_5/2_ and Mo 3d_3/2_ of Mo^6+^ [22]. The Ag 3d spectrum shown in Figure 2c can be divided into four characteristic peaks, in which the peaks at 368.8 and 374.3 eV are attributed to Ag 3d_5/2_ and Ag 3d_3/2_ of metallic Ag, meanwhile the peaks at 368.0 and 374.0 eV are assigned to Ag 3d_5/2_ and Ag 3d_3/2_ of Ag^+^, respectively. In addition, the peaks with I 3d_5/2_ and I 3d_3/2_ can also be observed at 619.9 and 631.4 eV in Figure 2d, respectively. Hence, the results from XPS clearly demonstrated the successful preparation of Ag/AgI/α-MoO_3_ with heterojunctions.

The morphology and microstructure of as-made pure α-MoO_3_ nanobelts and AAM-35 heterojunctions were investigated by SEM and (HR)TEM measurements, and the results are shown in Figure 3a–f. For pure *α*-MoO_3_, a unique belt-shaped morphology with smooth surface, sharp top and uniform belt width of about 100 nm is observed (Figure 3a). Crystalline orthorhombic *α*-MoO_3_ nanobelts with *Pbnm* symmetry was confirmed by (HR)TEM images (Figure 3b,c), the lattice fringe corresponded to (110) plane of the *α*-MoO_3_ (JCPDS No. 05-0508), and is clearly visible with a spacing of 0.379 nm. For Ag/AgI/MoO_3_ heterojunction (AAM-35), Figure 3d and e clearly show that Ag/AgI nanoparticles with the size of 20–30 nm located on the surface of the *α*-MoO_3_ nanobelts form different Ag/AgI, AgI/MoO_3_ and Ag/MoO_3_ heterostructures. In order to confirm the existence of metallic Ag in AAM-35, HRETM image shown in Figure 3f indicated three obviously different lattice fringes that respectively belong to the (111) plane with a spacing of 0.369 nm for AgI with *F-43m* symmetry, the (040) reflection plane with a spacing of 0.342 nm for *α*-MoO_3_, and the (111) planes with spacing of 0.232 nm for face-centered cubic metallic Ag nanoparticles with *Fm-3m* symmetry. These lattice fringes clearly reveal the formation of a ternary heterojunction, Ag/AgI/*α*-MoO_3_, which probably further promotes the effective separation of electrons and holes between the Ag, AgI and *α*-MoO_3_.

The UV-Vis DRS spectra of pure α-MoO_3_ nanobelts, AgI, and AAM heterojunctions are shown in Figure 4a. The absorption edge of pure α-MoO_3_ and AgI was estimated to be about 420 and 458 nm, respectively. However, owing to the formation of heterojunction structures among AgI, Ag and α-MoO_3_, the absorption edge of the AAM nanocomposite exhibited an obvious red-shift phenomenon accompanied by a gradually enhancing intensity with an increasing initially formed AgI amount. This is according to the following equation, [F(R_∞_)*hγ*] =A*(hγ-E_g_)^n/2^*, where F(R_∞_), *γ*, *E_g_*, *A* and *h* are the Kubelka–Munk function, the light frequency, the band gap, the proportionality constant, and Planck constant, respectively. In this equation, n depends on whether the transition is direct (n = 1) or indirect (n = 4) semiconductor type. For α-MoO_3_ and AgI, n = 1 [36]. The band gap of pure α-MoO_3_ nanobelts and AgI is calculated to be 3.06 and 2.78 eV, respectively. For Ag/AgI/MoO_3_ composite, the profile of (*αhγ*)^2^ versus *hγ* is shown, but the band gap of composite cannot be simply calculated using this method. Moreover, for pure α-MoO_3_ nanobelts and AgI, the valance band (VB) potential (*E*_VB_) and the conduction band (CB) potential (E_CB_) can be calculated by the empirical equations:*E*_VB_ = χ − *E*_e_ + 0.5*E*_g_
*E*_CB_ = *E*_VB_ − *E*_g_,
where χ is the absolute electronegativity of the semiconductor, which is the geometric mean of the electronegativity of the constituent atoms. *E*_e_ is the energy of free electrons on the hydrogen scale (about 4.5 eV vs. NHE) and *E*_g_ is the band gap of the semiconductor. The values of χ for pure α-MoO_3_ nanobelts and AgI are 6.40 and 5.40 eV, respectively [47]. As a consequence, the *E*_VB_ of pure α-MoO_3_ nanobelts and AgI are 3.43 and 2.29 eV, respectively, and the *E*_CB_ of pure α-MoO_3_ nanobelts and AgI are 0.37 and −0.49 eV, respectively.

Photoluminescence (PL) spectrum and electrochemical impedance spectroscopy (EIS) are widely used to monitor the separation and transfer efficiencies of the photogenerated charge carrier. For the PL spectrum, the high emission intensity implies a high recombination rate of the photogenerated charge carrier [48]. For pure α-MoO_3_ nanobelts, AgI, and AAM-35, the PL emission intensity of AAM-35 depicted in Figure 5a was weaker than that of pure α-MoO_3_ nanobelts and AgI, indicating that the modification of Ag/AgI on the surface of α-MoO_3_ nanobelts efficiently improved the separation of photogenerated e^−^ and h^+^. Moreover, the arc size of the AAM-35 from the EIS results was smaller than that of the *α*-MoO_3_ nanobelt and AgI, further suggesting that AAM-35 possessed smaller charge-transfer resistance is in favor of photogenerated charge carrier transfer. This result is consistent with the PL property of AAM-35 [49].

### 3.2. Photocatalytic Oxidative Desulfurization Activity

Thiophene is a refractory sulfur-containing pollutant in gasoline, which is chosen as a model of sulfur-containing pollutant to evaluate the photocatalytic oxidative desulfurization (PODS) activity of as-made photocatalysts in the AAM series. As can be seen in Figure 6a, the pure α-MoO_3_ nanobelt and AgI as the photocatalysts show the degradation rate of thiophene of 34.5% and 75.4% under visible-light irradiation for 2 h, respectively. However, under an identical condition, the AAM series exhibited superior PODS activity, in which AAM-35 had a highest PODS activity with a thiophene degradation rate of 97.5%. Note that the PODS activity of AAM series gradually increased with the increasing loading of the initially formed AgI amount, and then decreased. In our studies, 35 wt % AgI in all starting chemicals was the formed starting amount and is an inflexion point in AAM series. The excess loading of Ag/AgI nanoparticles occupied the partially active sites of surface of α-MoO_3_ nanobelts and thereby reduced the efficiency of the charge carrier separation [49]. As a comparison, when AgI/α-MoO_3_-35 (AM-35) was used as a photocatalyst for the degradation of thiophene, the degradation rate reached 87.6%. This result is obviously lower than that of AAM-35, confirming that Ag nanoparticles coated on an AgI or α-MoO_3_ surface act as a key role for the efficient enhancement of the PODS activity of AAM-35. These results also indirectly imply that Ag nanoparticles are beneficial to the improvement of electron-hole separation efficiency. Roles of AgI and Ag/AgI are in good agreement with that of AgI/Bi_2_SiO_5_ for the degradation of acid red aqueous solution (ARG) and gaseous formaldehyde [39] and AgI/Ag/Bi_2_MoO_6_ for the degradation of organic pollutants [40].

During photocatalysis, the photocatalyst amount used in the reaction system is an important factor to affect the conversion of substrate. Herein, we investigated the amounts of AAM-35 ranging from 0.5 to 2.0 g·L^−1^ for the influence of the photocatalytic degradation of thiophene under visible-light irradiation. As shown in Figure 6b, the thiophene degradation speed initially increased and then decreased with increasing amount of catalyst. When the concentration of catalyst was 1.0 g L^−1^, the degradation of thiophene could be completed in 2 h. If it was less or higher than this concentration, thiophene degradation was markedly weakened due to insufficient catalyst or that the excessive amount of photocatalysts blocked the light penetration [50].

To obtain a further insight of the photocatalytic process, the experimental results were simulated by pseudo-first-order kinetic model, and the results are shown in Figure 6c. The photodegradation process of thiophene on various catalysts fitted well with a pseudo-first-order model, ln (*C_0_*/*C_t_*) = *k*_app_t, where *k*_app_ is the apparent rate constant (min^−1^). *C_0_* and *C_t_* are the thiophene concentrations at corresponding reaction time 0 and t, respectively. For thiophene degradation, the results demonstrated that with increasing irradiation time the plot of ln (*C_0_*/*C_t_*) versus irradiation time (t) is a typical pseudo first-order reaction in the presence of photocatalysts. The corresponding rate constants (*k*_app_) for pure α-MoO_3_, AgI, AM-35, AAM-15, AAM-25, AAM-35 and AAM-45 are 3.21 × 10^−3^, 8.49 × 10^−3^, 2.04 × 10^−2^, 1.28 × 10^−2^, 2.94 × 10^−2^, 4.89 × 10^−2^ and 1.49 × 10^−2^ min^−1^, respectively. The *k*_app_ of AAM-35 is the highest, which is 15.2, 5.8 and 2.4 times higher than that of pure α-MoO_3_, AgI and AM-35, respectively. These findings reveal that Z-scheme Ag/AgI/α-MoO_3_ heterojunction-structured catalysts can significantly improve the photocatalytic activity for the degradation of thiophene compared to single α-MoO_3_ and AgI, and dual composite AgI/MoO_3_. Moreover, photocatalytic performances of all catalysts are in good agreement with the PL and EIS results.

The stability and recyclability of photocatalysts are often used to evaluate their practical applications in industry. Herein, catalyst AAM-35 was run four times for thiophene degradation. The results are shown in Figure 7a. It is clear to see that the catalytic activity of catalyst slightly decreased from 97.5% to 86.2% after four cycles, which could be ascribed to the reduction of the active sites and loss of the active component [49]. The XRD patterns and UV-DRS spectra before and after four times confirm that no remarkable changes can be observed (Figure 7b,c), although the UV-DRS spectrum shows a similar trend with a slight change (Figure 7c). These results are enough to corroborate the high stability and recyclability of Z-scheme Ag/AgI/α-MoO_3_ heterojunction-structured catalyst, which can be a promising photocatalyst in the practical application for PODS in the future.

### 3.3. Photocatalytic Mechanism

To elucidate and understand the reaction mechanism, the active species trapping experiments were carried out. Ethylenediaminetetraacetic acid disodium salt (EDTA-2Na), isopropanol (IPA), and benzoquinone (BQ) were adopted as the scavengers for photogenerated holes (h^+^), hydroxyl radical (OH), and superoxide radical (O_2_^−^), respectively. The experimental results shown in Figure 8 reveal that the addition of EDTA-2Na and IPA markedly suppresses the PODS activity of AAM-35. When the EDTA-2Na and IPA were added into the reaction system, the thiophene degradation rates declined from 97.5% to 18.3% and 42.4%, respectively, suggesting that h^+^ and ·OH should be the mainly active species, wherein h^+^ played a greater role than OH during photocatalysis. Moreover, the addition of BQ slightly weakened the thiophene degradation, implying that O_2_^−^ had also a small contribution to the thiophene degradation. Hence, for AAM-catalyzed thiophene degradation, the h^+^ and OH are two primary active species and the O_2_^−^ is secondary active species in the PODS process.

To further identify the radical species in the reaction process, the spin trapping ESR spectra utilized DMPO (5,5-Dimethyl-1-pyrroline N-oxide) as a trapping agent for hydroxyl radicals (·OH) and superoxide anionic radicals (·O_2_^−^); these were measured under visible light irradiation in the presence of pure α-MoO_3_ nanobelt, or AgI, or AAM-35. As can be seen in Figure 9, for all catalysts, no any radical signals can be observed in the dark. However, under visible light irradiation, the concentration of hydroxyl radicals is much higher than ·O_2_^−^ for pure α-MoO_3_ nanobelts (Figure 9a,b), indicating that the primary active species are the radicals ·OH. For AgI, the signals of DMPO-OH radicals are much weaker than ·O_2_^−^ (Figure 9c,d), corroborating that the radicals·O_2_^−^ are absolutely dominant species, which are derived from the reduction of physical adsorbed O_2_ on AgI surface by the photogenerated electron at the CB of AgI. For AAM-35, note that the signals of both of radicals ·O_2_^−^ and ·OH are observed and their intensities increase with increasing irradiation time (Figure 9e,f), implying the co-existence of two primary active species ·O_2_^−^ and ·OH. These findings are consistent with the results of active species trapping experiments, further verifying that the photocatalytic process was performed by the synergistic effect of active species h^+^, ·OH and·O_2_^−^.

On the basis of the above-mentioned experimental results, the photocatalytic mechanisms of AM (AgI/α-MoO_3_) and AAM (Ag/AgI/α-MoO_3_) are proposed and schematically illustrated in Scheme 2. Under visible light irradiation, both α-MoO_3_ nanobelts and AgI could be excited to generate electron-hole pairs (e^−^-h^+^), as shown in Scheme 2a, for AM heterojunction, due to the conduction band potential of α-MoO_3_ being much higher than that of AgI and the valence band potential of AgI being much lower than that of α-MoO_3_, and therefore, the electrons at CB of AgI can readily transfer to the CB of α-MoO_3_, meanwhile the holes from VB of α-MoO_3_ nanobelts can transfer to VB of AgI. Such transfer leads to the effectively spatial separation of photo-generated electron-hole pairs on AM catalyst and therefore performs the reducing reaction of O_2_ to ·O_2_^−^ on AgI surface and the conversion of H_2_O to radicals ·OH on α-MoO_3_ nanobelts. These radicals or holes further interact with thiophene and decompose it to form the final CO_2_, SO_3_ and H_2_O. As a result, the separation efficiency of photogenerated charge carriers of AM-35 are significantly enhanced compared to single α-MoO_3_ and AgI, due to the synergistic effect of heterojuction structure of dual composites. Note that AM-35 can partially convert into a similar structure as AAM-35 during the visible-light irradiation-catalyzed process. A similar case of AgI partially converted into metallic Ag under visible light irradiation had been reported previously on photocatalyst Bi_2_SiO_5_/AgI for the degradation of acid red G aqueous solution [51].

However, when partial AgI in AM-35 was reduced to metallic Ag until the AgI surface was completely encapsulated by metallic Ag under visible light irradiation to form Ag/AgI and Ag/α-MoO_3_ heterojunctions, while accompanied by the preservation of initial AgI/α-MoO_3_ heterojunction, then multiply heterojunction-structured Ag/AgI/α-MoO_3_ (AAM-35) was prepared in this case. In the AAM-35 heterojunction system, except from the above-mentioned AM-35-catalyzed photocatalytic mechanism, under visible-light irradiation, photo-induced electrons and holes on MoO_3_ and AgI were transferred by different routes. Due to the E_CB_ of MoO_3_ being more negative than the Fermi level of metallic Ag, photo-generated electrons on MoO_3_ would like to move to metallic Ag. Because the E_VB_ of AgI is more positive than the Fermi level of metallic Ag, electrons in Ag transferred to the VB of AgI, while photo-induced holes on the surface of AgI would spontaneously transfer to metallic Ag and combined with the transferred electrons on the VB of AgI, which was faster than the recombination between electrons and holes of AgI itself [52]. Thus, the efficient charge transmission could enhance charge carriers’ separation efficiency. This case occurs at the interface MoO_3_/Ag/AgI heterojunction structure. Additionally, the transfer of holes on AgI can also happen at interface AgI/Ag [39]. The resulting holes (h^+^) are richened on MoO_3_ to perform oxidation reaction—the holes at the VB of α-MoO_3_ can interact with thiophene and H_2_O to form radical C_4_H_4_S^+^ cation and ·OH due to the fact that the *E_VB_* of α-MoO_3_ (3.44 eV vs. NHE) is more positive than the standard redox potential ·OH/OH^−^ (+2.40 eV vs. NHE), and the electrons are richened on AgI to carry out reduction reactions—the electrons at the CB of AgI reduced the adsorbed O_2_ on the surface of heterojunctions to ·O_2_^−^. The resultant active species further react with C_4_H_4_S^+^ to produce final SO_3_ and CO_2_ to perform the effective degradation of thiophene [7]. These pathways greatly improve the separation efficiency of photo-induced electrons and holes and thereby remarkably promote photocatalytic performance of catalyst. To identify the final product of the AAM-catalyzed photodegradation of thiophene, 0.1 M Ba(NO_3_)_2_ aqueous solution was introduced to the reaction system. After finishing the photoreaction, the white precipitates were collected by centrifugation and characterized by XRD. The results shown in Appendix A confirm that white precipitates are the mixture of BaSO_4_, BaCO_3_ and AAM-35, suggesting that the final products of thiophene photodegradation catalyzed by AAM-35 are SO_3_ and CO_2_. This result is in good agreement with that of previous reports catalyzed by Ag/MoO_3_ [46] and Pd/ZrO_2_–chitosan composite [5]. In summary, the metal Ag in the Z-scheme heterojunction as a photogenerated charge carriers transfer bridge could significantly promote the electrons and holes’ transmission and separation, resulting in better stability and superior photocatalytic activity.

## 4. Conclusions

In this study, a series of novel Z-scheme photocatalysts Ag/AgI/α-MoO_3_ (AAM) heterojunctions with initially formed AgI contents were successfully fabricated by a facile hydrothermal method combined with a charge-induced physical adsorption and photo-reduced deposition technique, and well-characterized by XRD, SEM/TEM analysis, XPS and spectroscopies. The results revealed that AgI and Ag derived from the reduction of AgI were incorporated to α-MoO_3_ or AgI surface to form different heterojunctions AgI/α-MoO_3_, Ag/AgI and Ag/α-MoO_3_ for dual/ternary composites AM-35/AAM-35. For the photocatalytic oxidative desulfurization of thiophene, ternary composites AAM-35 (Ag/AgI/α-MoO_3_) heterojunction exhibited the highest photocatalytic activity (the conversion of 97.5% in 2 h) under visible-light irradiation compared to the single composite pure α-MoO_3_ nanobelts and AgI, and dual composites AM-35 (AgI/α-MoO_3_). Two different catalytic mechanisms were suggested to elucidate the effective separation of photogenerated electron-hole pairs for the enhancement of photocatalytic performance for dual composites AM-35 and ternary composites AAM-35 during the photocatalytic oxidative desulfurization of thiophene, revealing a key role of Ag and the importance of heterojunctions in a multiple-composite system. For the PODS of thiophene, three dominantly active species confirmed by radical trapping experiments and ESR spectra are h^+^, ·OH and·O_2_^−^. The final products of PODS of thiophene are SO_3_, CO_2_ and H_2_O verified by the formation of BaSO_4_ and BaCO_3_ in the presence of Ba(NO_3_)_2_. This investigation demonstrates that PODS is a promising candidate for refractory sulfur aromatic pollutant’s removal in fuel, also implying that the fabrication of Z-scheme-type heterojunctions might be a new approach for achieving the environmental requirements of VLD photocatalysts. However, how to regulate photocatalyst interface performance so that it can be easily separated from the reaction system, which should be addressed in the future.

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
