# Peer review of "The Novel Z-Scheme Ternary-Component Ag/AgI/α-MoO3 Catalyst with Excellent Visible-Light Photocatalytic Oxidative Desulfurization Performance for Model Fuel"

_nanomaterials, 2019, doi:10.3390/nano9071054_

Round 1
Reviewer 1 Report
This paper did systematic research on the preparation and characterization of Ag/AgI/α-MoO3 photocatalyst designed for oxidative desulfurization of fuels. The obtained materials were characterized by many techniques and finally the probable photocatalytic mechanism was proposed and discussed. In reviewer's opinion, this valuable work is recommended for publication after minor revision.
1. In the title and in other parts of the text Authors used the form “photocatalytic oxidation desulfurization”. The word “oxidation” should be rather replaced by “oxidative”. Therefore the phrase “photocatalytic oxidative desulfurization” is more correct.
2. In the description of UV-Vis spectra used for determination of band gap energies, the function on the Y-axis in the plot (b) should be expressed as a modified Kubelka-Munk function.
3. The discussion on the proposed Z-scheme mechanism (Scheme 2) needs some clarification. The role of metallic silver as a charge carriers bridge should be more clearly explained and better indicated in the Scheme 2b.
Reviewer 2 Report
The authors report the synthesis of a novel ternary system of Ag/AgI/MnO3 (AAM) photocatalysts with highly efficient desulfurization under visible-light irradiation. Several characterizing techniques were carried out to investigate their properties. Solid evidence and clear discussion were also provided to confirm the enhancement of photocatalytic activity for AAM-35. I think this works is interesting and will be useful for the researchers in the field of photocatalysts. However, the authors should address the following comments before I recommend for publication in Nanomaterials.
1. The authors mentioned “AAM-35” in the abstract without the description of 35 meaning before. The author should denote the meaning of 35 in the abstract.
2. In Figure 1, “JCDPF” should be revised to “JCPDS”.
3. In Figure 3f, I recommend the authors to draw the lines to indicate the regions corresponding to Ag, AgI and MnO3.
4. The photocatalytic efficiency of AAM-35 decreased from 97.5% to 86.2% after 4 cycles. Can the authors explain more about the possible reasons why the photocatalytic efficiency decreased? How about the leaching of Ag nanoparticles from MnO3 supports?
5. The use of photocatalysts in powder form for desulfurization in the fuel may be difficult to implement in the practical industrial process owing to the difficulty in collection of powder after use. Do the authors have any suggestion for further development of this AAM catalyst in practical use?
